# Genotype-by-Environment Interaction Stability Analysis of New Quinoa (*Chenopodium quinoa* Willd.) Varieties in the Mediterranean Zone of Chile

**DOI:** 10.3390/plants14193007

**Published:** 2025-09-28

**Authors:** Pablo Olguín, Samuel Contreras, Claudia Rojas, Francisco Fuentes

**Affiliations:** 1Facultad de Agronomía y Recursos Naturales, Pontificia Universidad Católica de Chile, Santiago 8331150, Chile; paolguinm@uc.cl (P.O.); scontree@uc.cl (S.C.); carojas12@uc.cl (C.R.); 2Facultad de Ingeniería, Pontificia Universidad Católica de Chile, Santiago 8331150, Chile; 3Facultad de Medicina, Pontificia Universidad Católica de Chile, Santiago 8331150, Chile

**Keywords:** multi-environment, breeding, correlation, sea level ecotype, AMMI-Biplot

## Abstract

Quinoa (*Chenopodium quinoa* Willd.), a crop native to the Andean region, exhibits variable performance in yield components under rainfed Mediterranean conditions. Consequently, identifying varieties that demonstrate stability in key agronomic traits—regardless of environmental fluctuations—is essential for enhancing crop reliability and productivity. In this work, new five varieties belonging to the sea-level ecotype (Pangal, Nieves, Pincoya, Chucao and Regalona), with superior performance to local materials used by farmers in terms of uniformity, stability, yield characteristics, grain diameter, thousand-grain weight, protein percentage, and saponins, were established in three environments (Pichilemu 34°29′ S/72°01′ W, Coihueco 36°42′ S/72°42′ W, Cañete 37°51′ S/73°24′ W) during two consecutive seasons (2019/2020, 2020/2021). Data analysis confirmed narrow variability among the varieties analyzed and between season and environment in all characteristics evaluated. The Pichilemu area (close to the coast) was the most productive over the two years of cultivation studied, with grain yields reaching 2975 kg·ha^−1^. In the Cañete (coastal) and Coihueco (foothill) environments, grain yields were 2892 and 2453 kg·ha^−1^, respectively. The Pangal variety (pearl) had the highest grain yield, reaching 3162 kg·ha^−1^ in all environments. Nieves (white) variety had the best grain diameter (GD = 1.88 mm) and the best thousand-grain weight (TGW = 3.10 g). Regarding grain protein concentration, the Pincoya (black) variety had the highest score (GP = 16.31%). The lowest concentration of Saponin was obtained in Chucao (red) variety (GS = 1.46%). The Additive Main Effects and Multiplicative Interaction (AMMI) analysis did not identify any variety that exhibited greater yield and stability. Consequently, over the two years of study, the Nieves and Pangal varieties presented the best yield in the Pichilemu environment, with 3673 and 3788 kg·ha^−1^, respectively. These varieties also stood out in the Cañete environment as obtaining the best yields (3547 and 3169 kg·ha^−1^); however, they did not obtain the highest yield in the Coihueco environment. The Chucao variety was considered to have greater stability obtaining average yield in all study environments. This study presents a comprehensive phenotypic characterization of newly developed varieties, offering insights into their adaptive relationships with Mediterranean environments. To further elucidate the influence of environmental stressors on agronomic performance and grain quality traits, future trials are recommended in more extreme ecological settings.

## 1. Introduction

Quinoa (*Chenopodium quinoa* Willd.) is increasingly recognized for its agronomic versatility and nutritional value—characteristics that have fueled its global expansion [1]. Its grain is rich in protein, contains a balanced amino acid profile, and harbors beneficial compounds such as saponins and flavonoids [2]. In addition to its nutritional appeal, quinoa’s tolerance to abiotic stresses including drought, salinity, and temperature extremes makes it a viable crop for sustainable agriculture under shifting climatic regimes [3].

However, the transition of quinoa cultivation to Mediterranean regions presents specific constraints [4]. Rainfed farming systems dominate these areas, characterized by winter rainfall, dry summers, and high thermal amplitude [5]. These conditions—compounded by photoperiod sensitivity and longer growth cycles of traditional Andean varieties—limit crop establishment and yield stability [6]. Consequently, local breeding efforts have pivoted toward identifying and developing germplasm with better adaptation to temperate latitudes and water-limited environments [7].

The genetic diversity of quinoa is structured into distinct ecotypes—highland, salar, valley, Yungas, and lowland—each adapted to specific environmental niches [8]. Of these, the coastal lowland ecotype, cultivated below 600 m.a.s.l., is of particular relevance for Mediterranean agriculture [9]. Its short growth cycle, heat tolerance, and adaptability under limited water availability offer promising traits for developing high-performing, stable varieties suited to rainfed cultivation [10]. In Chile, this ecotype has become a focal resource for breeding programs aiming to deliver varieties with improved grain size, protein content, saponin reduction, and overall field performance [11].

To accurately select genotypes with desirable traits and environmental responsiveness, multi-environment trials (METs) are essential [12]. These trials allow breeders to evaluate genotype-by-environment (G × E) interactions, which play a crucial role in determining yield and quality stability across spatial and temporal gradients [13,14,15,16]. A seminal study by Bertero et al. (2004) evaluated 24 quinoa cultivars across 14 diverse environments and found that G × E interaction effects were four times more influential than genotype effects for grain yield and similarly significant for grain size [12]. These findings established the importance of ecotype-specific breeding strategies, emphasizing that no single genotype performs uniformly across all environments [17].

Subsequent studies have expanded on these insights. For instance, Thiam et al. (2021) demonstrated that G × E effects accounted for over 30% of yield variability in Moroccan trials, reinforcing the ecological relevance of origin-based adaptation [18]. Nguyen et al. (2024) explored quinoa performance under waterlogging and drought, revealing genotypes with strong resilience to early-stage stress [19]. Meanwhile, Souri Laki et al. (2025) employed multiple stability indices to rank Bolivian and Peruvian genotype sizes in Iranian environments, underscoring the need for multi-trait selection [20]. Adaptation studies in tropical zones (Nacchio-Jojoa et al., 2023) further confirm that trait expression varies significantly by location, and that genotype recommendations must be contextual [21].

To statistically interpret G × E interactions, the Additive Main Effects and Multiplicative Interaction (AMMI) model has proven to be a robust analytical tool [22,23]. It combines analysis of variance (ANOVA) for genotype and environment effects with principal component analysis (PCA) for the interaction term, allowing researchers to partition variability and visualize genotype performance across environments [14]. AMMI has been successfully applied in major staple crops—including rice, maize, and wheat—and its increasing use in quinoa breeding programs provides a structured approach for identifying varieties with broad or specific adaptability [14,24].

This study aims to analyze the productivity and environmental stability of new quinoa varieties recently released in Chile using AMMI analysis, based on agronomic traits such as yield, grain size, thousand-grain weight, and nutritional composition across three Mediterranean environments over two consecutive seasons.

## 2. Results

### 2.1. Environmental and Seasonal Variation in Productive Variables

Analysis of variance showing F and *p* values for year, genotype, environment, and interactions among the five characteristics that were evaluated presented different magnitudes of statistical significance. For the year (Y) parameter, results were significant for almost all the study variables, except thousand-grain weight. For genotypes (G), all the study variables showed significance. Environment (E) showed significance in most of the variables, with the exception of grain diameter (Table 1). For the parameters Y × E, Y × G, G × E and Y × E × G, significance was observed for yield, protein content, and saponin (Table 1). The Y × E parameters were significant for grain diameter and thousand-grain weight.

The analysis of variance showed high variability in yield characteristics among varieties across both crop years (Table 2). Yield ranged from 1300 to 4344 kg·ha^−1^, grain diameter from 1.53 to 2.09 mm, and thousand-grain weight from 1.59 to 4.28 g. Yields by variety, considering the environments, ranged from 2453 to 2975 kg·ha^−1^ in Coihueco and Pichilemu, respectively (Table 3). Regarding grain diameter, no significant differences were observed between environments, with an average of 1.79 mm (Table 3). Thousand-grain weight ranged from 1.59 g in Coihueco to 2.92 g in Pichilemu.

### 2.2. Environmental and Seasonal Variation in Grain Quality Variables

Regarding grain protein content, the average ranges varied from 14.27 to 16.31% in the Regalona and Pincoya varieties, respectively. The highest record (19.44%) was identified in the Coihueco environment, in the Pincoya variety. Saponin levels ranged from 0.9 to 4.1% in the Chucao variety (Pichilemu environment) and the Pincoya variety (Coihueco environment), respectively. The Chucao variety stood out for having the lowest average grain saponin content, at 1.46% (Table 2).

Regarding grain saponin content, the average for all varieties was 2.03%. No variety was considered sweet or had less than 1.4% saponin. The Chucao variety had the lowest saponin content across all environments (1.46%) (Table 2). The Nieves variety had the highest saponin content, at 2.73%. No significant differences were observed in grain saponin content between environments. However, the highest saponin content was found in the Cañete environment (2.14%), followed by Coihueco (2.05%) and Pichilemu (1.91%) (Table 3).

### 2.3. Cluster Analysis

The cluster analysis (Figure 1A,B) revealed biologically meaningful groupings among both genotypes and environments, which align with the patterns of adaptation and stability observed in the AMMI analysis. In Figure 1A, genotypes clustered into three distinct groups based on agronomic and nutritional traits. Thus, the first group (G1) was composed of the Nieves and Regalona varieties, characterized by a white and cream (pearly) grain color, with yields of 2724 and 2824 kg·ha^−1^ and grain diameters of 1.84 and 1.88 mm, indicating potential for direct consumption and broader environmental plasticity. Thousand-grain weights were 3.10 and 3.94 g and the protein contents of these varieties were very close, at 14.27 and 14.29 g, respectively, reflecting a balance between nutrient use efficiency and the nutritional quality of these varieties. However, for the saponin content variable, a greater distance was identified, from 2.32% for Nieves to 2.73% for Regalona, which could be related to differentiated defense mechanisms against herbivores or environmental stress, as saponins protect plants, but, in high concentrations, affect the palatability and processing of grains.

The second group (G2) was composed of the varieties Pangal and Chucao, of a yellow and red color, respectively. They showed more heterogeneous characteristics, with yields of 2842 and 3162 kg·ha^−1^, grain diameters of 1.75 and 1.86 mm, and thousand-grain weights of 2.57 and 3.05 g, respectively. They are indicative of a slight variation in the density and development of the endosperm, possibly related to growing conditions. Protein content varied between 14.41 and 14.70% and saponin content from 1.46 to 1.70%, showing consistent grain quality, although slightly lower than that of G1. This could be linked to biological or agronomic selection favoring less bitter varieties that are better suited for direct human consumption. These differences could reflect specific adaptations to local environments, where factors such as solar radiation, temperature, or nutrient availability influence grain quality and yield. The results also showed a separated cluster of the Pincoya variety, characterized by black grain, a yield of 2.314 kg·ha^−1^, a grain diameter of 1.62 mm, a thousand-grain weight of 2.19 g, and a protein content of 16.31%, which could be the result of a distinct adaptation strategy that is possibly suited to more limiting conditions. (Figure 1A).

The classification of the environments for all study variables generated two equidistant groups (Figure 1B). The grouping discriminates between environments with low altitude (50–100 m.a.s.l.) that are close to the coastline (G1) with moderate temperatures ranging from 9.7 °C (September) to 15.7 °C (February) and a relative humidity of 78% during the growing season (September–March). However, in the Pichilemu environment, rainfall during the growing season ranged from 0.0 to 37.6 mm per month, while the Cañete environment had a range of 0.7 to 114.5 mm. The second group (G2) was in the Coihueco locality, with a higher altitude (650 m.a.s.l.), greater thermal oscillation, average temperatures of 9.7 °C in September and 20.2 °C in January, a precipitation range of 0.1 to 114.5 mm per month, and a humidity of 62% (Figure 4). These environmental clusters correspond with genotype performance patterns in the AMMI biplots, reinforcing the genotype–environment interaction structure and supporting the identification of stable and specifically adapted varieties.

### 2.4. AMMI—Biplot Model

The AMMI biplots (Figure 2 and Figure 3) provide a detailed visualization of genotype–environment interactions for key agronomic traits. The first principal component axis (PC1) explained the majority of the interaction variance—86.9% for yield, 96.8% for grain diameter, and 86.3% for thousand-grain weight—indicating that most of the interaction signal is captured in the horizontal dimension of the biplots.

The Biplot graph in Figure 2A shows the results regarding yield variable, with the Cañete and Pichilemu environments presenting higher yields than the Coihueco environment. The Chucao variety exhibited high stability across environments. On the other hand, Pangal and Nieves varieties showed greater interaction effects and specific adaptation to coastal sites (Cañete and Pichilemu environments). The second axis (PC2), although accounting for a smaller proportion of the variance, helped differentiate environments such as Coihueco, which presented distinct climatic conditions and contributed to genotype performance divergence (Figure 2B). These patterns reinforce the clustering results and support a nuanced understanding of varietal adaptability, guiding selection strategies for both broad and specific environmental targets.

For the grain diameter variable (Figure 2C), Cañete environment was found to be the location in which the largest grain diameter was obtained for all varieties. The strongest association was observed between the Nieves variety and Cañete environment. As shown in Figure 2D, the Nieves variety presented with a greater grain diameter in all environments and was also identified as the variety with the greatest stability with respect to this variable. Contrary to what was observed in the Chucao and Pincoya varieties, these had smaller grain diameters in Coihueco compared to the Pichilemu environment. The Biplot graph in Figure 2E for the thousand-grain weight variable identified the Pichilemu environment as having the greatest stability. The Pangal and Chucao varieties were associated with the Pichilemu environment. The Regalona variety was linked to the Coihueco environment; however, the Pincoya and Nieves varieties were not fully associated with any environment. Figure 2F shows that the Chucao variety had the highest stability across all environments, despite not having the highest grain weight. The other varieties had a dissimilar response with respect to environment, and less stability was observed for this variable.

In relation to grain protein content (Figure 3A), a pattern of higher percentages for the Pincoya and Pangal varieties (>15%) was observed in low-altitude environments near sea level (Pichilemu and Cañete). A second pattern observed for all three varieties was associated with a lower percentage, approximately 14.5%. The Pincoya variety, which had the highest grain protein concentration, was not associated with any environment. The greatest stability in all environments for the grain protein trait (Figure 3B) was observed for the Chucao variety, followed by the Regalona variety; however, these were not the varieties with the highest protein concentrations. The Pincoya variety showed a slight decrease in its content in the Cañete environment, but greater distance from the other varieties. The Nieves and Pangal varieties showed the greatest instability with respect to all environments.

With respect to the saponin contents of grain shown in Figure 3C, no specific environment was identified as having lower saponin content among all varieties. For this characteristic, most of the varieties showed no association with any environment, with the exception of the Pincoya variety, which obtained the highest saponin concentration in the Coihueco environment (Figure 3C). In Figure 3D, the Chucao variety was identified as having the lowest saponin content in all environments. Nieves, Regalona, and Pangal showed lower saponin content in the Coihueco environment and higher saponin content in the Pichilemu environment. The opposite was observed for the Pincoya variety, whose saponin content was lowest in the Pichilemu environment (Figure 3D).

## 3. Discussion

The global promotion of the International Year of Quinoa by the FAO (Food and Agriculture Organization of the United Nations) during 2013 generated a steady increase in quinoa consumption [25]. Several countries, mainly European, generated a demand for this grain, which is mainly obtained from Peru and Bolivia [26]. Currently, there is a new requirement in Europe for superior quinoa varieties, with high yields, a larger grain size, and high protein content, which must be adaptable to European environmental conditions [9]. In this new scenario, genetic material from the coastal or lowland ecotypes are projected as essential for the development of new varieties with the capacity to adapt and generate yields greater than 2 ton·ha^−1^ in Mediterranean agricultural environments [5]. Coastal or lowland ecotypes are emerging as essential for developing varieties capable of achieving yields exceeding 2 tons per hectare in Mediterranean environments.

Our study aimed to identify the grain yield and quality components of four new commercial varieties, along with the Regalona variety, which all belong to the coastal ecotype. Regalona has been widely used as a global comparison standard [27]. However, it had not previously been evaluated against new varieties from the coastal ecotype and those with a distinct grain color in the context of three different environments in the Mediterranean zone of central Chile.

The study by Bertero et al. (2004) [12] examined the GXE analysis model in an international context. It considered genotypes and varieties from the Andean region, as well as from Europe, and analyzed them in various environments, including the extreme conditions of the Chilean desert environment (Tarapacá Region). The effects of genotype and environment interactions were evaluated to measure the stability of the varieties under different environmental conditions. Data analysis showed high variability among the genotypes/varieties analyzed and between season and environment for all the characteristics evaluated. A study of adaptation of new quinoa genotypes to temperate zones showed a significant interaction between genotypes and environments for all the characteristics investigated, probably due to differences in climatic conditions over the years [12].

Several studies have revealed a growing interest in studying the effect of irrigation to ensure high and stable yields in different environments. It has also been emphasized that good agricultural practices can ensure quinoa production in marginal and water-stressed environments [17]. In this study, the environments where the trials were established showed differences in precipitation, temperature and humidity: for the Pichilemu environment, in a normal season, there is no significant rainfall during the growing season, in contrast to the Coihueco and Cañete environments, where precipitation is present during all the growing months. Considering this, it is important to mention some studies which mention that Chilean germplasm is more tolerant to high temperatures than Andean germplasm [3,28]. The association generated in the cluster analysis identified the Pichilemu and Cañete environments with greater proximity, leaving the Coihueco environment as a separate cluster. This is due to the higher temperature conditions during the growing season, coupled with higher precipitation, which gives this environment unique climatic conditions.

Temperature and humidity records were comparable between the two years. The average humidity was 76.9% in Pichilemu, 63.6% in Coihueco, and 73.9% in Cañete. Some studies indicate that the high relative humidity under high temperature conditions could reduce plant transpiration [3,29]. The precipitation was much higher in the first year 2019/2020 in Cañete (227.5 mm) than in 2020/2021 (129.6 mm), while the opposite occurred in the Pichilemu and Coihueco environments, where the second year, 2020/2021, received higher precipitation during crop development, from 12.2 to 56.4 mm in Pichilemu and from 97.8 to 184.7 mm in the Coihueco environment, concentrated during the final stage of grain filling, between the end of February and the beginning of March. It is worth mentioning that, during the 2020/2021 season, unusual rainfall was recorded at the end of January, with more than 34 mm falling in less than 24 h in Pichilemu, and more than 100 mm in Coihueco, a climatic event that did not affect the development of the varieties established in the experimental units.

The new varieties are emerging as essential for achieving yields exceeding 2 tons per hectare. Yield also increased during the second year of cultivation in Pichilemu and Coihueco, from 2640 and 2274 kg·ha^−1^ in 2019/2020 to 2633 and 3311 in 2020/2021 kg·ha^−1^, respectively. These results are consistent with those of Emrani et al. [6], which identified a positive effect on the yield of all the study varieties in the year of the trial where precipitation increased during grain filling. However, one of the important characteristics of the new varieties is their adaptability to low-rainfall conditions during crop development [30]. This new type of genetic material is vital for quinoa’s adaptation and stability to variable climatic conditions, making these varieties a key tool for agriculture in Mediterranean environments. According to Del Pozo et al. [31], a set of genotypes were identified that show a high stress tolerance index, and therefore greater tolerance to drought conditions, which are candidates for future research focused on the selection of parents for more advanced reproductive stages [12,32].

In terms of grain quality, all genotypes with protein contents exceeding 14.27% were identified, with the Pincoya variety standing out, which recorded an average of 16.31% in the three test environments. This is an important characteristic when projecting these new varieties as commercial varieties with low fertilization requirements that are adapted to the diversity of environments in Mediterranean areas, since quinoa production in rainfed Chile receives the remaining fertilizers in the soil due to the rotation with cereal crops [11,27,31,33]. Research measured grain protein content in various genotypes, using different doses of urea—35 kg N ha^−1^, 70 kg N ha^−1^, and 0 kg N·ha^−1^—obtained as averages and providing a total protein concentration in grain of 13.05%, 14.83%, and 16.7%, respectively [34]. These concentrations respond to similar measurements to our study, which did not use fertilization in the two years of the trial.

The production of saponin-free quinoa varieties has been an important breeding goal in recent years [35]. For breeding and phenotyping purposes, it is desirable to use a simple screening test, such as the aphrosymmetric method [36], which takes advantage of the foaming characteristics of saponins. The results of our study identified the Pichilemu environment (near the coast) as having the lowest average saponin content across the two growing seasons (1.91%), as opposed to the Coihueco environment, which had the highest saponin content (2.60%). Studies identify that grain saponin content can be influenced by various factors, such as location, abiotic stress, and variety. Factors such as drought and salinity decrease the amount of saponins [37]. For our study, we can preliminarily determine that environments near the coast, with higher humidity and more moderate temperatures, tend to lead to lower saponin content in the grain [36]. When analyzing the individual varieties, Chucao was identified, followed by Pangal, as having the lowest saponin content, at 1.46% and 1.70%, respectively. Studies on the extraction and quantification of saponins in quinoa have shown variability in this characteristic depending on the varieties evaluated. The principal components of AMMI comprised over 86% for all the study traits, which captures the largest proportion of the total variability in the analyzed data [38].

The study by Bertero et al., 2004 [12], provided a reference for the implications of G × E interactions for the genetic improvement of yield and grain size in quinoa in low-latitude environments. They assessed four genotype types (medium-altitude valleys, Northern plateau, Southern plateau, and sea level). Our study evaluated only varieties of the sea-level ecotype in three different environments and two agricultural seasons. To this end, the estimation of the effects of the G × E interaction in this work was carried out using the AMMI model. It was determined that the Chucao variety exhibited the most stable yield, while the Chucao and Pangal varieties obtained the highest results for the thousand-grain weight variable. The Pangal variety recorded the highest thousand-grain weight, with an average of 3.05 g over the two years and a maximum of 4.28 g recorded in the Pichilemu area. The yield and grain quality components of the new commercial varieties highlight their relevance as a global standard in different areas of the Mediterranean region. In agreement with Thiam et al. [18], grain yield and thousand-grain weight are the main variables that were positively correlated. In our study, the differences in thousand-grain weight were significant among varieties, with a highest mean value of 3.42 g and a higher mean for all varieties in the Pichilemu area. This study [18], like ours, was influenced by an exceptional rainy season, where the varieties reached their highest yield and thousand-grain weight.

For the protein content variable, the most stable varieties in the AMMI model were Chucao, Regalona, and Pincoya. For the variables of grain diameter and saponin content, no outstanding varieties were identified in terms of stability. This could be related to improved agronomic stability, a characteristic that the AMMI model can help quantify by analyzing genotype–environment interactions. As observed by Marques et al. [39], the use of the AMMI method in maize grain showed that, in terms of G × E interaction, it was identified that stability and adaptability were not always associated with higher yield; only 2 of the 36 maize genotypes were recommended for the central Brazilian environment due to their high adaptability and stability combined with higher yield. Therefore, the G × E model allows for the identification of new varieties with adaptability to a wide variety of environments to minimize crop losses [24]. Similar results were reported in spring barley: with G × E analysis using the AMMI method, soil fertility proved to be the main driver for yield and thousand-grain weight in the varieties analyzed [14]. In our study, no exceptional varieties were identified as having the ability to maintain high levels of yield and high grain quality in all environments: all varieties had yields higher than 2.3 ton·ha^−1^ (Table 3). The AMMI model was instrumental in estimating the adaptation and stability of quinoa to variable climatic conditions, becoming a key tool for the improvement of new varieties. By grouping environments with similar responses, the AMMI model helps reduce the number of trials required, optimizing resources and time in breeding programs. Overall, lowland or coastal ecotypes provide an opportunity to find genetic material with high productive yield and adaptability to the diversity of environments of the Mediterranean zone in Chile. This research is the result of a breeding program that made four new varieties of quinoa available. The variations in the productive characteristics with respect to the environments of the trials support the productive recommendations for each new variety in Mediterranean areas.

## 4. Materials and Methods

### 4.1. Study Areas and Experimental Design

Five quinoa varieties were evaluated in three environments in south–central Chile during the 2019/2020 and 2020/2021 seasons through multi-environment trials located in the localities of Pichilemu (34°29′16.43″ S/72°1′12.95″ W) in the O’Higgins region, Coihueco (36°42′43.94″ S/71°42′8.88″ W) in the Ñuble region, and Cañete (37°51′20.63″ S/73°24′37.93″ W) in the Bio-Bio region. Average rainfall in the growing season was 34.3, 141.2, and 178.5 mm in Pichilemu, Coihueco, and Cañete, respectively.

Four new varieties were identified from the genetic breeding program of Pontificia Universidad Católica de Chile, registered in 2023: Chucao, Nieves, Pincoya, and Pangal, registered in the Plant Variety Protection Registration Office of Chile. These were developed using a combination of Individual Selection and Stratified Mass Selection methods. Regalona Baer is an old variety, owned by the Campex Baer company in Chile.

Climatic data were collected from the meteorological stations closest to each analysis site (Figure 4). The recording of climatic variables during two growing seasons considered the following variables: maximum, minimum, and average temperatures (°C), precipitation (mm), humidity (%), and wind speed (k/h) (https://agrometeorologia.cl/) [12].

**Figure 4 plants-14-03007-f004:**
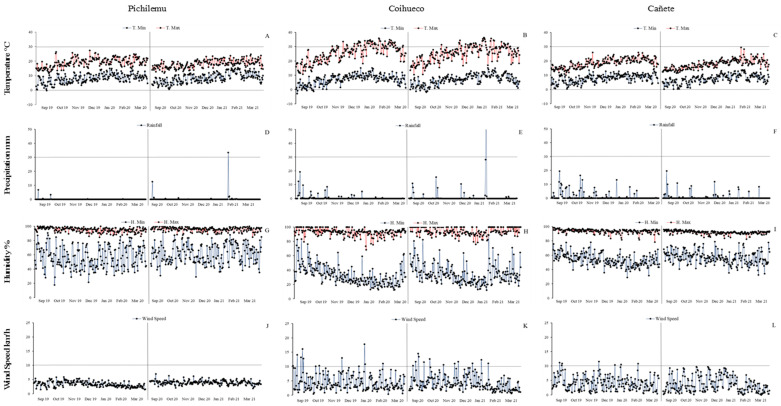
Temperature (°C) (**A**–**C**), Rainfall (mm) (**D**–**F**), humidity (%) (**G**–**I**), and wind speed (km/h) (**J**–**L**) between planting and harvest in Pichilemu, Coihueco, and Cañete for the 2019/2020 and 2020/2021 seasons.

Field trials were conducted in active agricultural production areas integrated within cereal–legume crop rotation systems. The study considered a completely randomized block design with three replications for the five varieties in the three environments. The experimental unit had a size of 9.6 m^2^ (2.4 × 4 m), composed of four furrows of 4 m length and an internal spacing of 0.6 m. Sowing was carried out with a manual push seeder, with a seeding density of 10 kg × ha^−1^ and a seeding depth of 2 cm [40].

Sowing took place in all environments at the beginning of September during both seasons. Harvesting was carried out when plants attained around 50% physiological grain maturity, which typically occurred 130 to 150 days post-sowing [41]. Plants were harvested manually in the center of each experimental unit, and once the plant material was separated from the grain, the variables grain yield (kg·ha^−1^), grain diameter (mm), and thousand-grain weight (g) were evaluated.

Crop management included soil preparation using a harrow and disk plow. Throughout crop development, conventional pest and disease management practices were implemented, tailored to local agroecological conditions and aimed at minimizing the impact of harmful agents on plant health. Selective phytosanitary products were applied to preserve ecological balance while promoting crop vigor (e.g., Karate^®^ Zeon 050 cs). Plots were systematically monitored to detect early signs of infestation or disease, enabling timely and targeted interventions. Weed control was conducted regularly using manual monitoring cards, ensuring effective suppression of weed growth and minimizing its interference with crop performance.

### 4.2. Grain Quality Analysis

To identify grain protein, first the N concentration (%) was determined for each dry and milled grain sample, using a LECO CN 628 automated dry combustion analyzer. Subsequently, nitrogen use efficiency was calculated using the nitrogen to protein conversion factor, applying the value 6.25, which is used for most plant proteins [42].

The saponin content in the grains of each sample was evaluated using the afrosimetric method, which is based on the ability of saponins to decrease the surface tension of water, forming a stable foam whose height is related to the content of saponins in the grains [43].

### 4.3. Statistical Analysis, AMMI Model

To identify superior genotypes for the three different environments in two growing seasons, different ANOVA tests were performed for yield, grain diameter, thousand-grain weight, protein content and saponin. Data normality was assessed using the Kolmogorov–Smirnov test, which confirmed that the data were normally distributed. Significant differences between treatments were estimated using Tukey’s multiple comparison test with *p* = 0.05. These analyses were performed using InfoStat statistical software-version 2008 [44].

To identify homogeneous groups among the five genotypes according to the variables analyzed, a hierarchical cluster analysis was performed [45]. This same model was also applied to identify groups in the study environments. To identify similarities in the performance pattern between genotypes and environment, a dendrogram was constructed [12].

The analysis of the AMMI model was performed using the metan package in the RStudio program, version 1.3.959 [30], applying the standard ANOVA model and principal component analysis (PCA), to quantify stability. The AMMI decomposition is based on the following statistical model:*Y_ger_* = *μ* + *α_g_* + *β_e_* + ∑ λ*_n_ γ_gn_ δ_en_* + *ρ_ge_* + *ε_ger_*
where *Y_ger_* represents the yield of genotype *g* in environment *e* for replicate *r*; *μ* represents the highest mean; *α_g_* and *β_e_* represent the genotypes and environments that deviate from *μ*; λ*_n_* represents the *n*th singular value of the principal component of the interaction (IPC); *γ_gn_* and *δ_en_* are the eigenvector values of genotype *g* and environment *e* of component *n*; *ρ_ge_* represents the residual of the AMMI model; and *ε_ger_* represents the error. Fixed effects correspond to the main effects of genotypes and environments, while random effects refer to the genotype–environment interaction [13,46]. The results of the AMMI analysis were presented using Biplot plots for each of the variables (yield, grain diameter, thousand-grain weight, and protein and saponin contents), as well as a Biplot plot containing all the analysis variables over the two years.

## 5. Conclusions

The varieties evaluated in the present study showed dissimilar ranges of variability in yield, grain diameter and weight, protein content, and grain saponin, as well as different grain colors (white, cream (pearl), yellow, black, and red). The G × E analysis developed for these varieties generates a useful tool for decision making regarding the selection of new desirable and stable varieties in breeding programs. This study is the first to analyze a subset of new varieties across different climates in Chile’s Mediterranean region, highlighting their potential for cultivation in other Mediterranean areas. The AMMI model allowed the effects of G × E to be described more precisely: the Chucao variety was widely adapted to diverse environments in the Mediterranean zone, presenting high yields, grain weights, and protein contents, thus minimizing crop losses generated by adverse environments and their maximization of them based on genetic factors. In addition, the Pincoya variety obtained the lowest yield (2.3 t/ha) among the varieties but had the highest level of grain proteins (16.31%). Thus, these new varieties are an attractive material for producers located in the Mediterranean regions, as well as making an important contribution of vegetable protein for the human diet.

This study presents a comprehensive phenotypic characterization of newly developed varieties, offering insights into their adaptive relationships with Mediterranean environments. To further elucidate the influence of environmental stressors on agronomic performance and grain quality traits, future trials are recommended in more extreme ecological settings.

## Figures and Tables

**Figure 1 plants-14-03007-f001:**
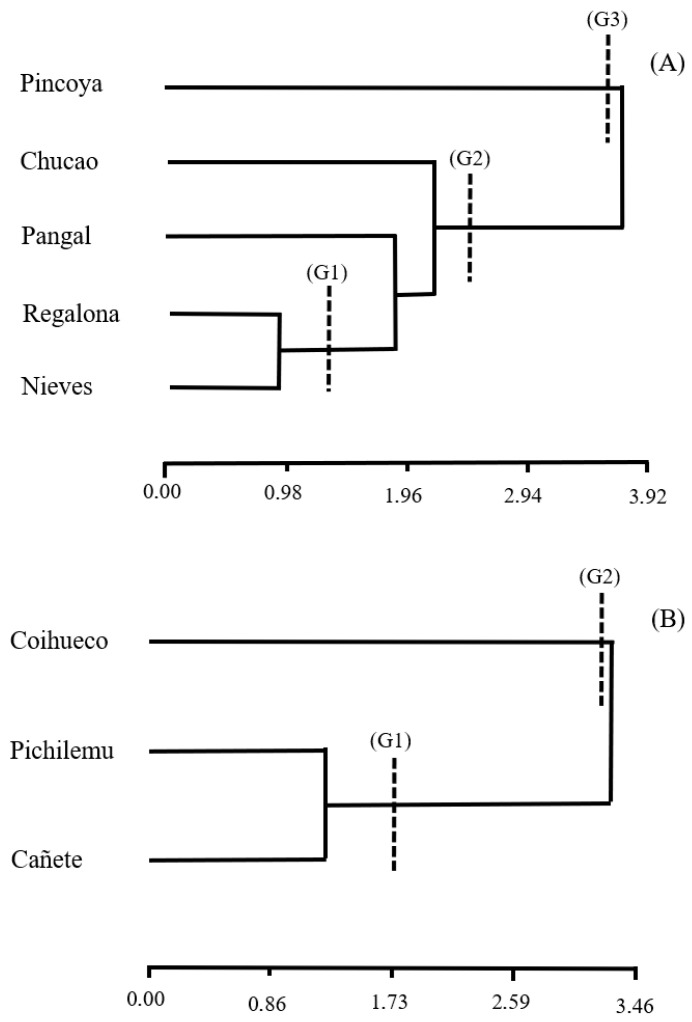
Cluster analysis of quinoa varieties, according to environmental and seasonal variations in productive and grain quality variables (**A**) and a cluster analysis of environments according to varietal and seasonal variations (**B**).

**Figure 2 plants-14-03007-f002:**
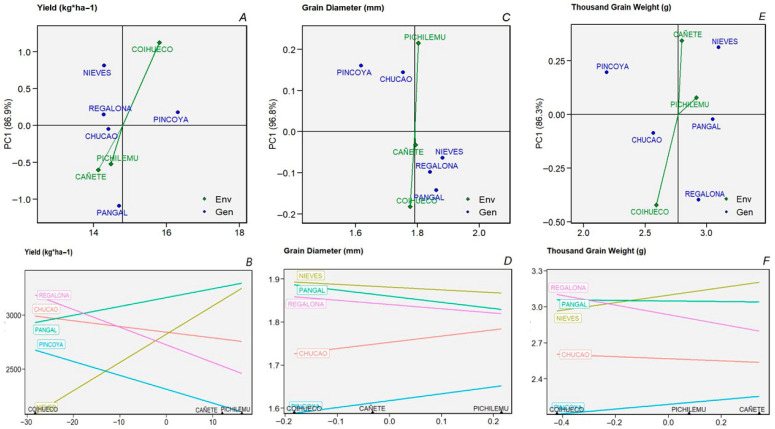
Differential response of genotypes in different environments, using AMMI stability analysis - Biplot type 2 (top graphs) and Biplot type 4 (bottom graphs) for yield (**A**,**B**), grain diameter (**C**,**D**) and thousand grain weight (**E**,**F**), for five varieties in three environments and two growing seasons.

**Figure 3 plants-14-03007-f003:**
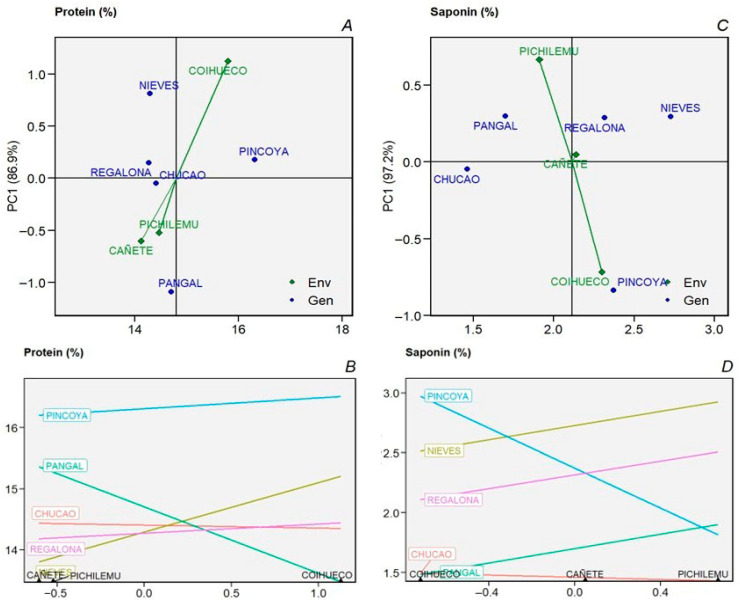
Differential response of genotypes in different environments, using AMMI stability analysis-Biplot type 2 (top graphs) and Biplot type 4 (bottom graphs) for protein (**A**,**B**) and saponin (**C**,**D**), for five varieties in three environments and two growing seasons.

**Table 1 plants-14-03007-t001:** Analysis of variance showing coefficient of variance F and probability of the statistical *p* value for year, genotype, environment, and interactions among the five evaluated characteristics.

Parameter	Yield (kg·ha^−1^)		Diameter Grains (mm)		Thousand-Grain Weight (g)		Protein (%)		Saponin (%)	
	*F*	*p*	*F*	*p*	*F*	*p*	*F*	*p*	*F*	*p*
Year	71.07 *	*p* < 0.001	65.89 *	*p* < 0.001	3.13 ^ns^	0.082	62.81 *	*p* < 0.001	0.01 *	*p* < 0.001
Genotype	18.27 *	*p* < 0.001	63.56 *	*p* < 0.001	38.64 *	*p* < 0.001	64.28 *	*p* < 0.001	32.09 *	*p* < 0.001
Environment	25.70 *	*p* < 0.001	1.60 ^ns^	0.210	11.74 *	*p* < 0.001	112.45 *	*p* < 0.001	2.76 *	*p* < 0.001
Year × environment	2.12 *	*p* < 0.001	21.01 *	*p* < 0.001	20.43 *	*p* < 0.001	154.15 *	*p* < 0.001	2.73 *	*p* < 0.001
Year × genotype	1.14 *	*p* < 0.001	0.92 ^ns^	0.459	1.52 ^ns^	0.208	12.29 *	*p* < 0.001	2.65 *	*p* < 0.001
Genotype × environment	12.95 *	*p* < 0.001	1.51 ^ns^	0.173	1.15 ^ns^	0.342	15.01 *	*p* < 0.001	1.62 *	*p* < 0.001
Year × environment × genotype	1.81 *	*p* < 0.001	1.10 ^ns^	0.379	1.72 ^ns^	0.111	10.56 *	*p* < 0.001	2.36 *	*p* < 0.001

* Significant at the 0.001 probability level; ^ns^, not significant at *p* = 0.05.

**Table 2 plants-14-03007-t002:** Mean, standard deviation, minimum, maximum, and coefficient of variation for the five characteristics of the five varieties, over two years of evaluation in three environments.

	Yield (kg·ha^−1^)					Diameter Grain (mm)					Thousand-Grain Weight (g)					Protein (%)					Saponin (%)				
Varieties	Mean	S.D.	Min	Max	C.V.	Mean	S.D.	Min	Max	C.V.	Mean	S.D.	Min	Max	C.V.	Mean	S.D.	Min	Max	C.V.	Mean	S.D.	Min	Max	C.V.
Nieves	2824 ab	888.52	1300	4133	31.46	1.88 a	0.09	1.67	2.01	4.91	3.10 a	0.41	2.19	3.73	13.14	14.29 b	1.83	11.88	18.00	12.82	2.73 a	0.53	1.60	4.10	19.47
Pangal	3162 a	610.32	2100	4344	19.30	1.86 a	0.10	1.68	2.09	5.46	3.05 a	0.43	2.51	4.28	14.10	14.70 b	1.54	12.31	16.63	10.50	1.70 bc	0.33	0.90	2.30	19.46
Pincoya	2314 c	310.10	1833	2889	13.70	1.62 c	0.10	1.53	1.93	6.11	2.19 c	0.42	1.59	3.41	19.29	16.31 a	1.53	13.75	19.44	9.39	1.95 a	0.68	1.00	3.20	28.76
Chucao	2842 ab	478.76	2167	3950	16.84	1.75 b	0.08	1.67	1.97	4.35	2.57 b	0.25	2.30	3.28	9.90	14.41 b	1.33	12.19	16.19	9.25	1.46 c	0.37	0.90	2.30	25.30
Regalona	2724 b	422.94	1933	3478	15.52	1.84 a	0.06	1.72	1.99	3.38	2.94 a	0.20	2.71	3.41	6.95	14.27 b	1.16	12.13	16.06	8.13	2.32 ab	0.32	1.80	2.70	13.67
Mean	2773		1866	3758		1.79		1.65	1.99		2.77		2.26	3.62		14.79		12.55	17.26		2.03		1.24	2.92	

Means not sharing any letter are significantly different according to Fisher’s test at the 5% level of significance.

**Table 3 plants-14-03007-t003:** Mean, standard deviation, minimum, maximum, and coefficient of variation for the five characters, in three environments, in all varieties evaluated over two years.

	Yield (kg·ha^−1^)					Diameter Grain (mm)					Thousand-Grain Weight (g)					Protein (%)					Saponin (%)				
Environment	Mean	S.D.	Min	Max	C.V.	Mean	S.D.	Min	Max	C.V.	Mean	S.D.	Min	Max	C.V.	Mean	S.D.	Min	Max	C.V.	Mean	S.D.	Min	Max	C.V.
Pichilemu	2975 a	718.30	1833	4344	24.14	1.80 a	0.15	1.53	2.09	8.22	2.92 a	0.57	1.96	4.28	19.42	14.47 b	1.36	12.50	16.94	19.42	1.91 a	0.67	0.90	3.30	35.09
Coihueco	2453 b	518.91	1300	3233	21.15	1.78 a	0.12	1.54	1.96	6.90	2.59 b	0.43	1.59	3.17	16.59	15.80 a	1.50	12.31	19.44	16.59	2.05 a	1.73	1.30	4.10	28.66
Cañete	2892 a	512.48	2000	3833	17.72	1.79 a	0.12	1.53	2.01	6.66	2.80 ab	0.42	1.84	3.73	15.12	14.12 b	1.64	11.88	17.25	15.12	2.14 a	0.60	1.00	3.20	28.01
Mean	2773		1711	3803		1.79		1.53	2.02		2.77		1.79	3.72		14.79		12.23	17.87		2.03		1.06	3.53	

Means not sharing any letter are significantly different according to Fisher’s test at the 5% level of significance.

## Data Availability

The original contributions presented in this study are included in the article. Further inquiries can be directed to the corresponding author.

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
