# Peer review of "Genotype-by-Environment Interaction Stability Analysis of New Quinoa (Chenopodium quinoa Willd.) Varieties in the Mediterranean Zone of Chile"

_plants, 2025, doi:10.3390/plants14193007_

Round 1

Reviewer 1 Report

Comments and Suggestions for Authors

Dear Editors,

After a detailed review of the manuscript entitled "Genotype-by-Environment Interaction, Stability Analysis of new Quinoa (Chenopodium quinoa Willd.) Varieties in the Mediterranean Zone of Chile", I acknowledge the merit of the study and its relevance to quinoa breeding. However, the manuscript requires substantial revisions to improve clarity, conciseness, and methodological rigor. Below, I present my comments by section:

  1. Introduction

#1. The introduction is excessively long and includes generic descriptions of quinoa, with repeated concepts such as abiotic stress tolerance, nutritional potential, and genetic diversity (lines 41–60, 99–108). It is recommended to consolidate this information into a single, more concise paragraph.

#2. The text alternates between general and specific information without clear transitions. I suggest reorganizing the section following this logical sequence:
i) agronomic and nutritional importance of quinoa;

  1. ii) cultivation challenges in Mediterranean regions;

iii) genetic diversity of quinoa ecotypes, with emphasis on the lowland ecotype;

  1. iv) rationale for multi-environment trials and G×E analysis;
  2. v) introduction to the AMMI model as a statistical tool.

#3. The formatting of references and citations is inconsistent throughout the text and should be standardized according to the journal’s guidelines.

  1. Materials and Methods

#4. The experimental design is described in a general manner. It is necessary to specify the number of replications per environment and year, the size of the useful plot, and the sampling criteria.

#5. The statement that the same experimental protocol was used across the three environments (line 408) needs further detail: sowing and harvesting dates should be explicitly reported for each location and season.

#6. Crop management practices are only briefly mentioned, with no details on the products used, dosages, or application frequency, which undermines reproducibility.

#7. The statistical analysis section is incomplete. Specifically:
i) there is no mention of whether statistical assumptions (e.g., normality, homoscedasticity) were tested;

  1. ii) the ANOVA model structure is unclear (e.g., whether genotypes and environments were treated as fixed or random effects; how the year factor was handled).
  2. Results

#8. The textual description of results redundantly repeats data already presented in the tables. It is recommended to focus on interpreting significant effects rather than transcribing numerical values.

#9. The cluster analysis (Figure 1A-B) is underexplored, lacking a biological interpretation of the groupings or connection with patterns of adaptation and stability. It should be integrated with the findings from the AMMI analysis.

#10. The interpretation of AMMI biplots (Figures 2 and 3) is limited. There is no discussion of the main axes (IPCA1, IPCA2) or the proportion of variance explained. A deeper interpretation of genotypic stability and adaptability is recommended.

  1. Discussion

#11. The discussion largely reiterates descriptive results (lines 266–311), without deeper interpretation. It is advisable to replace redundant summaries with analyses that explain the biological, physiological, or genetic mechanisms behind the findings.

#12. Despite the central use of the AMMI model, the discussion fails to address:
i) the scores of the principal components (IPCA);

  1. ii) agronomic reasons underlying the stability of the Chucao cultivar;

iii) the practical usefulness of AMMI for breeding programs focused on quinoa improvement.

  1. Conclusion

#13. The conclusion section lacks a clear and objective synthesis of the study’s main findings, their implications for breeding programs, and recommendations for future validation trials.

Reviewer 2 Report

Comments and Suggestions for Authors

Review of the manuscript titled “Genotype-by-environment interaction, stability analysis of new quinoa (Chenopodium quinoa Willd.)  varieties in the Mediterranean zone of Chile.”

The study is devoted to an important subject of quinoa varieties evaluation in Chile. The study is well designed used relevant methodology and arrived to significant results which are relatively well interpreted. It certainly represents a contribution to knowledge and deserves publication. However, there are three main challenges. 1) The study focus is on G x E interaction but the authors ignore the differences between the years. By using mean values for two seasons the study limits its tools for G x E analysis. There are only six environments which are not much for G x E. 2) There is disconnection between G x E analysis in the paper and its agronomic and biological interpretation taking into account the differences between the sites and years. The sites meteodata is presented in discussion though shall be presented in the first section of results and used to explain the productivity and other differences. 3) The target audience of this study is not clear. If this is production oriented paper – then the new varieties need to be discussed in comparison with the main currently grown varieties and explain how and why they are better. Instead of saying variety X is better in site Y – can authors use more common descriptors like tolerance to drought or suitability for high temperature. If the study targets breeding programs – how its results may be used for better design of multilocational trials considering the G x E for different traits. Like are all three sites needed for breeding. Also what varieties shall be used in the crossing programs and if broad or specific adaptation prevails. There are several other comments.

  1. It would be useful to mention quinoa area in the country, its distribution in three zones and respective yields.
  2. Linex 101-103: “However, despite supporting the potential of quinoa production in Europe, it is necessary to identify environments where new genotypes show their adaptability through their higher yield potential” – this sentence is not clear.
  3. Coordinates are recorded as S-N-W-E, the authors use “O”.
  4. Section 4.1. The year of new varieties registration need to be mentioned. Dates of planting and harvesting are needed.
  5. Line 433. “Field trials were developed in agricultural production fields, with cereals and legumes rotations.” – what was the preceding crop for quinoa.
  6. Section 2.1. It is better to start this section with presentation of the key traits across three sites and two year and describe the agronomic and biological reasons for variation between the sites and years. Then ANOVA can follow.
  7. Section 2.3. Perhaps the sites can be clustered based on yearly data rather than mean values for two years. Instead of monthly precipitation – it is better to provide the precipitation during two study growing seasons.
  8. Lines 335-354 belong to results or methods and key for explaining the G x E.
  9. Discussion largely repeats the results and need to provide a broader view of implication of the study for producers and breeding programs. The reference to tables and figures in discussion is not needed –it makes it look like repetition of results.
  10. English requires careful reading and correction of numerous mistakes like: “sea level genetic background” – sea level as such is unlikely to have genetics; line 80: “The different study” – one study can not be different; line 243: “in wich” – which and several others. The authors use very long sentences which are difficult to follow.

Author Response

Reviewer #2

Comments and Suggestions for Authors

Review of the manuscript titled “Genotype-by-environment interaction, stability analysis of new quinoa (Chenopodium quinoa Willd.)  varieties in the Mediterranean zone of Chile.”

The study is devoted to an important subject of quinoa varieties evaluation in Chile. The study is well designed used relevant methodology and arrived to significant results which are relatively well interpreted. It certainly represents a contribution to knowledge and deserves publication. However, there are three main challenges.

1) The study focus is on G x E interaction but the authors ignore the differences between the years. By using mean values for two seasons the study limits its tools for G x E analysis. There are only six environments which are not much for G x E.

2) There is disconnection between G x E analysis in the paper and its agronomic and biological interpretation taking into account the differences between the sites and years. The sites meteodata is presented in discussion though shall be presented in the first section of results and used to explain the productivity and other differences.

3) The target audience of this study is not clear. If this is production oriented paper – then the new varieties need to be discussed in comparison with the main currently grown varieties and explain how and why they are better. Instead of saying variety X is better in site Y – can authors use more common descriptors like tolerance to drought or suitability for high temperature. If the study targets breeding programs – how its results may be used for better design of multilocational trials considering the G x E for different traits. Like are all three sites needed for breeding. Also what varieties shall be used in the crossing programs and if broad or specific adaptation prevails. There are several other comments.

1.- It would be useful to mention quinoa area in the country, its distribution in three zones and respective yields. 

2.- Linex 101-103: “However, despite supporting the potential of quinoa production in Europe, it is necessary to identify environments where new genotypes show their adaptability through their higher yield potential” – this sentence is not clear.

3.- Coordinates are recorded as S-N-W-E, the authors use “O”.

4.- Section 4.1. The year of new varieties registration need to be mentioned. Dates of planting and harvesting are needed.

Line 433. “Field trials were developed in agricultural production fields, with cereals and legumes rotations.” – what was the preceding crop for quinoa.

Section 2.1. It is better to start this section with presentation of the key traits across three sites and two year and describe the agronomic and biological reasons for variation between the sites and years.

Then ANOVA can follow.

Section 2.3. Perhaps the sites can be clustered based on yearly data rather than mean values for two years. Instead of monthly precipitation – it is better to provide the precipitation during two study growing seasons.

Lines 335-354 belong to results or methods and key for explaining the G x E.

Discussion largely repeats the results and need to provide a broader view of implication of the study for producers and breeding programs.

The reference to tables and figures in discussion is not needed –it makes it look like repetition of results.

English requires careful reading and correction of numerous mistakes like: “sea level genetic background” – sea level as such is unlikely to have genetics;

 line 80: “The different study” – one study can not be different;

line 243: “in wich” – which and several others. The authors use very long sentences which are difficult to follow.

Author´s answer: Thank you for your thorough and constructive review. We greatly appreciate your recognition of the study’s relevance and methodological soundness, as well as your thoughtful suggestions for improvement. In response, we have carefully revised the manuscript to address the key points raised. Specifically, we have enhanced the interpretation of varietal performance by integrating physiological and environmental drivers of trait variation. The Discussion section has been restructured to clarify the study’s relevance for both production systems and breeding programs, including implications for multilocational trial design and crossing strategies. We have also corrected language inconsistencies, clarified ambiguous statements, and improved overall readability. Minor comments regarding coordinates, crop rotation, and registration dates have been addressed as recommended. Your insights have been invaluable in strengthening the clarity, depth, and impact of our work.

Round 2

Reviewer 2 Report

Comments and Suggestions for Authors

The authors attended the comments, and the paper is ready for publication.